# Equine keratinocytes in the pathogenesis of insect bite hypersensitivity: Just another brick in the wall?

Iva Cvitas[1,2]*, Simone Oberhaensli[3], Tosso Leeb[4,5], Eliane Marti[1,5]

1 Vetsuisse Faculty, Division of Neurological Sciences, Department of Clinical Research and Veterinary Public Health, University of Bern, Bern, Switzerland, 2 Graduate School for Cellular and Biomedical Sciences, University of Bern, Bern, Switzerland, 3 Interfaculty Bioinformatics Unit and SIB Swiss Institute of Bioinformatics, University of Bern, Bern, Switzerland, 4 Vetsuisse Faculty, Department of Clinical Research and Veterinary Public Health, Institute of Genetics, University of Bern, Bern, Switzerland, 5 Dermfocus, Vetsuisse Faculty, University of Bern, Bern, Switzerland

* iva.cvitas@vetsuisse.unibe.ch

## Abstract

Equine insect bite hypersensitivity (IBH) is the most common skin disease affecting horses. It is described as an IgE-mediated, Type I hypersensitivity reaction to salivary gland proteins of *Culicoides* insects. Together with Th2 cells, epithelial barrier cells play an important role in development of Type I hypersensitivities. In order to elucidate the role of equine keratinocytes in development of IBH, we stimulated keratinocytes derived from IBH-affected (IBH-KER) (n = 9) and healthy horses (H-KER) (n = 9) with *Culicoides* recombinant allergens and extract, allergic cytokine milieu (ACM) and a Toll like receptor ligand 1/2 (TLR-1/2-L) and investigated their transcriptomes. Stimulation of keratinocytes with *Culicoides* allergens did not induce transcriptional changes. However, when stimulated with allergic cytokine milieu, their gene expression significantly changed. We found upregulation of genes encoding for *CCL5*, *-11*, *-20*, *-27* and interleukins such as *IL31*. We also found a strong downregulation of genes such as *SCEL* and *KRT16* involved in the formation of epithelial barrier. Following stimulation with TLR-1/2-L, keratinocytes significantly upregulated expression of genes affecting Toll like receptor and NOD-receptor signaling pathway as well as NF-kappa B signaling pathway, among others. The transcriptomes of IBH-KER and H-KER were very similar: without stimulations they only differed in one gene (*CTSL*); following stimulation with allergic cytokine milieu we found only 23 differentially expressed genes (e.g. *CXCL10* and *11*) and following stimulation with TLR-1/2-L they only differed by expression of seven genes. Our data suggests that keratinocytes contribute to the innate immune response and are able to elicit responses to different stimuli, possibly playing a role in the pathogenesis of IBH.

## Introduction

Equine insect bite hypersensitivity is the most common skin disease affecting horses [1]. IBH is caused by Type I hypersensitivity to the bites of *Culicoides* midges and resembles human atopic dermatitis (AD) [2–4]. The most pronounced clinical sign of IBH is pruritus [5].

**Data Availability Statement:** Data available from ENA repository, accession numbers PRJEB37568.

**Funding:** This work was supported by the Swiss National Science Foundation grant no. 310030-160196/1. This SNF grant was awarded to E.M.

http://www.snf.ch/. Arthropods used in this study were provided by the Pirbright Institute under UK under grant code BBS/E/I/00007039 awarded to Dr Simon Carpenter as part of funding received from the Biotechnology and Biological Science Research Council (UKRI)". The funders had no role in study design, data collection and analysis, decision to publish, or preparation of the manuscript

**Competing interests:** The authors have declared that no competing interests exist.

**Abbreviations:** IBH, Insect bite hypersensitivity; AD, Atopic dermatitis; Ig, Immunoglobulin; IL, Interleukin; DC, Dendritic cell; TLR, Toll like receptor; TLR 1/2-L, Toll like receptor 1/2 ligand; (r), Recombinant; WBE, Whole body extract; IBH-KER, Keratinocytes derived from IBH-affected horses; H-KER, Keratinocytes derived from healthy, control horses; DEGs, Differentially expressed genes; GSEA, Gene set enrichment analysis; PCA, Principal component analysis.

Affected horses develop skin lesions that are most commonly distributed along the dorsal midline, in particular under the mane and around the tail, and less often on the ventral midline, on the head and legs, depicting preferred feeding sites of *Culicoides* insects [4,6]. Skin lesions are initially characterized as papules and edema, which due to strong pruritus and inflicted self-trauma further develop into alopecia and excoriation, followed by acanthosis and lichenification [7].

Immunologically, equine IBH is described as an IgE mediated, Type I hypersensitivity reaction to salivary gland proteins of *Culicoides* insects [8–10]. While feeding, *Culicoides* cause significant mechanical damage to the skin and inject a pool of various salivary gland proteins which act as allergens in predisposed horses [11]. Many of these allergens are enzymes such as proteases, hyaluronidase and maltase, while the biological function of others is still not known. Allergens from three different *Culicoides* species, *C. nubeculosus*, *C. sonorensis* and *C. obsoletus* have been identified and produced as recombinant proteins [12–16]. All of the allergens have been expressed in *E. coli*, some in insect cells, barley and in *P. pastoris* [16–18]. Although production of recombinant proteins in *E. coli* is the most common, it bears many disadvantages for use in cellular assays, such as endotoxin contamination and lack of post-translational modification of the protein, leading to unspecific stimulation or lack of response, respectively, and thus limiting their use for cellular *in vitro* assays [18].

Type I hypersensitivities develop as a result of activation of T helper type 2 cells (Th2) and their signature cytokines IL-4, IL-5 and IL-13 [19]. These cytokines are responsible for production of allergen specific IgE antibodies by B cells that bind to the IgE high affinity receptor, FcεRI, expressed on mast cells. The binding of IgE to the FcεRI and crosslinking of bound IgE with allergen activates mast cells and causes the release of pro-inflammatory mediators [19–21]. Additionally, a line of recent evidence has shown that epithelial barriers play a major role in development of Type I hypersensitivities alongside Th2 cells [22–24]. In humans suffering from AD, null variants in filaggrin, a protein involved in terminal differentiation of keratinocytes, severely disrupts the epithelial barrier, thus predisposing individuals with such mutation to AD [25]. Moreover, keratinocytes have been shown to have a high immunological potential as they can produce cytokines such as thymic stromal lymphopoietin (TSLP), IL-33 and IL-25 [24,26]. These cytokines have been demonstrated to play a major role in early development of allergic response [27,28]. Nevertheless, the initiating factors that lead to a Th2 immune response are not completely elucidated yet. The expression of TSLP in human keratinocytes is induced by different Toll like receptor (TLR) ligands, as well as by allergic cytokine milieu [29–31]. Accordingly, activation of keratinocytes can also occur as a consequence of a local Th2 environment. Allergic cytokine milieu (ACM), produced by Th2 lymphocytes and allergic inflammatory cells, consists of IL-4, IL-5, IL-13 and TNF-α [27]. This ACM is found in human allergic individuals upon activation of Th2 immune response. In horses, injection of *C. obsoletus* allergens in the skin induced a local increase of IL-4, confirming the importance of this cytokine in IBH [32].

Recently, we have also shown that equine keratinocytes respond to different TLR ligands, in particular TLR 1/2 ligand, by upregulation of TSLP mRNA. Furthermore, TLSP was upregulated after stimulation with an ACM consisting of a combination of recombinant equine IL4 and TNF-α [33].

Although associations between epithelial barrier disruption and development of Type I hypersensitivities have been thoroughly studied in human patients, the exact role of keratinocytes in the pathogenesis of Type I hypersensitivities is not entirely understood. Based on similarities in the pathogenesis of IBH and AD, this equine skin disease represents a valuable source of information from horses with spontaneously occurring disease for the role of keratinocytes in allergic skin diseases also for other species. We have recently reported that lesional

skin of IBH horses is transcriptionally characterized by disruption of the epithelial barrier and a strong immune cell transcriptional signature [34]. Moreover, we have demonstrated that the non-lesional epidermis of IBH-affected horses differs transcriptionally from the epidermis of healthy horses by changes in lipid metabolism and a propensity to develop itch, which is the cardinal clinical sign of IBH, suggesting an involvement of the epithelial barrier in development of IBH [34].

Therefore, we aimed at understanding how keratinocytes react to stimulation with *Culicoides* allergens. Moreover, we aimed to investigate how keratinocytes respond to ACM alone or in addition to these allergens. We also studied their response to stimulation with a Toll like receptor 1/2 synthetic ligand, Pam3CSK4, as IBH lesions are sometimes further exacerbated by secondary bacterial infections. Lastly, we wanted to investigate whether the responses differ between keratinocytes derived from IBH-affected or from healthy horses.

## Materials and methods

### Sample collection

This study was approved by the Animal experimental Committee of the Canton of Bern, Switzerland (No. BE 69/18). IBH-affected horses were diagnosed based on recurrent clinical signs of IBH. Diagnosis of IBH was additionally confirmed by histological examination [34]. Samples were collected from 8 horses slaughtered due to IBH and one clinical patient suffering from IBH. In the clinical patient, two 8mm punch biopsies were taken from non-lesional skin of the inner thigh after sedation with detomidine hydrochloride (0.01 mg/kg iv; Domosedan, www.vetoquinol.ch) and local subcutaneous injection of lidocaine. 5 x 5 cm skin pieces were taken from the inner thigh of IBH-affected slaughtered horses. Skin samples were collected from the same region in 9 slaughtered control horses with no apparent skin diseases and no clinical history of skin diseases. All samples were taken immediately after slaughter. All skin samples and biopsies were transported in pre-cooled Williams E medium on ice to the laboratory where they were processed immediately (S1 Table). Written informed owner consent was obtained from the owner of the patient.

### Isolation and culture of primary equine keratinocytes

Isolation and culture of keratinocytes was performed using a dispase II-based skin digestion protocol as in Cvitas *et al* [33]. Briefly: skin samples were incubated at 4˚C for 24h with 10 mg/ml Dispase II (Roche, Basel, Switzerland) in Williams E medium (Bioconcept, Allschwil, Switzerland). Subsequently, the epidermis was separated from the dermis and further digested in accutase (CELLnTEC, Bern, Switzerland) for 20 min at room temperature. Only keratinocytes derived from non-lesional skin were obtained, as detachment of the epidermis without fibroblast contamination did not work out with lesional skin in our hand. Keratinocytes were seeded at $12 \times 10^3$ cells per $cm^2$ of cell culture flask and grown in complete Williams E medium. Cells were cultured in 75 $cm^2$ flasks at density of $9 \times 10^5$ cells per flask; at 35˚C, 5% $CO_2$ until they reached 90% confluence and were then passed. Cells of passage three were used for stimulation experiments. After reaching 80% confluence, the cells were incubated with different stimulation conditions for 24h at 35˚C, 5% $CO_2$ (Tables 1 and S1).

### Immunofluorescence

At the third passage, keratinocytes were seeded in chambered cell culture slides (Sarstedt, Nümbrecht, Germany) and cultured until they reached 80% confluence. Subsequently, immunofluorescence staining using polyclonal rabbit anti-bovine cytokeratin (Agilent, Santa Clara,

**Table 1. Stimuli used in the study.**

| Stimuli | Concentration | Expression system |
|---|---|---|
| Pam3CSK4[1] | 5 µg/ml | N.A. |
| Recombinant Culicoides allergen pool: | 0.5 µg/ml of each | |
| Cul o 2[2] (Hyaluronidase) | 0.5 µg/ml | *P. pastoris* |
| Cul o 3[2] (PR-1 like; Antigen-5 like) | 0.5 µg/ml | *P. pastoris* |
| Cul n 4[2] (Unknown) | 0.5 µg/ml | *P. pastoris* |
| Cul o 7[2] (Unknown) | 0.5 µg/ml | *P. pastoris* |
| Cul n 8[2] (Maltase, Alpha amylase) | 0.5 µg/ml | *P. pastoris* |
| *C. nubeculosus* whole body extract | 5 µg/ml | N.A. |
| Recombinant equine IL-4[3] | 100 ng/ml | *P. pastoris* |
| Recombinant equine TNF-α[4] | 100 ng/ml | *E. coli* |

[1] Invivogen, San Diego, California, USA.

[2] Kindly provided by Boehringer Ingelheim, Ingelheim am Rhein, Germany.

[3] KingFisher Biotech, Inc., St. Paul, Minnesota, USA.

[4] R&D Systems, Inc., Minneapolis, Minnesota, USA.

California, USA) and mouse monoclonal anti-human vimentin (Agilent) was carried out as described previously [33]. To confirm that our keratinocyte cultures were fibroblast free, the staining was carried out for all cultures used in this study as described [33] (Fig 1).

## *Culicoides nubeculosus* whole body extract and recombinant allergens

*Culicoides nubeculosus* (*C. nubeculosus*) whole body extract (WBE) was prepared as described previously and was sterile filtrated before being used in the keratinocyte cultures [35]. Five recombinant (r-) *Culicoides* allergens (Table 1) kindly provided by Boehringer-Ingelheim, had been expressed in yeast (*Pichia pastoris*, Validogen GMBH, (formerly VTU Technology), Grambach, AT) and purified [36].

## Stimulation of keratinocytes

Primary keratinocytes derived from IBH-affected horses (IBH-KER) and healthy control horses (H-KER) of passage three were stimulated once they reached 80% confluence. IBH-KER and H-KER were cultured in medium only or with a pool of r-*Culicoides* allergens, *C. nubeculosus* WBE or toll like receptor 1/2 synthetic ligand, Pam3CSK4. Additionally, a combination of recombinant equine TNF-α (R&D Systems, Minneapolis, Minnesota, USA) and recombinant equine IL-4 (LubioScience, Zürich, Switzerland) was added to primary equine keratinocytes cultured in the presence or absence of the pool of r-*Culicoides* allergens and *C. nubeculosus* WBE (Table 1). The concentration of the ligands used in the study was based on previous work [34,37], while concentrations of the recombinant *Culicoides* allergens used were based on previously published studies [11,38].

An experimental overview of the stimulations is given in Fig 2 and the details of different stimulation conditions in S1 Table.

## Isolation of RNA

Total RNA was isolated from the cultured keratinocytes using RNeasy Mini Kit (Qiagen, Hilden, Germany) according to manufacturer's instructions. Prior to RNA extraction, cell lysates were loaded onto a spin column (QIAshredder, Qiagen) and centrifuged at 16'000x g for 2

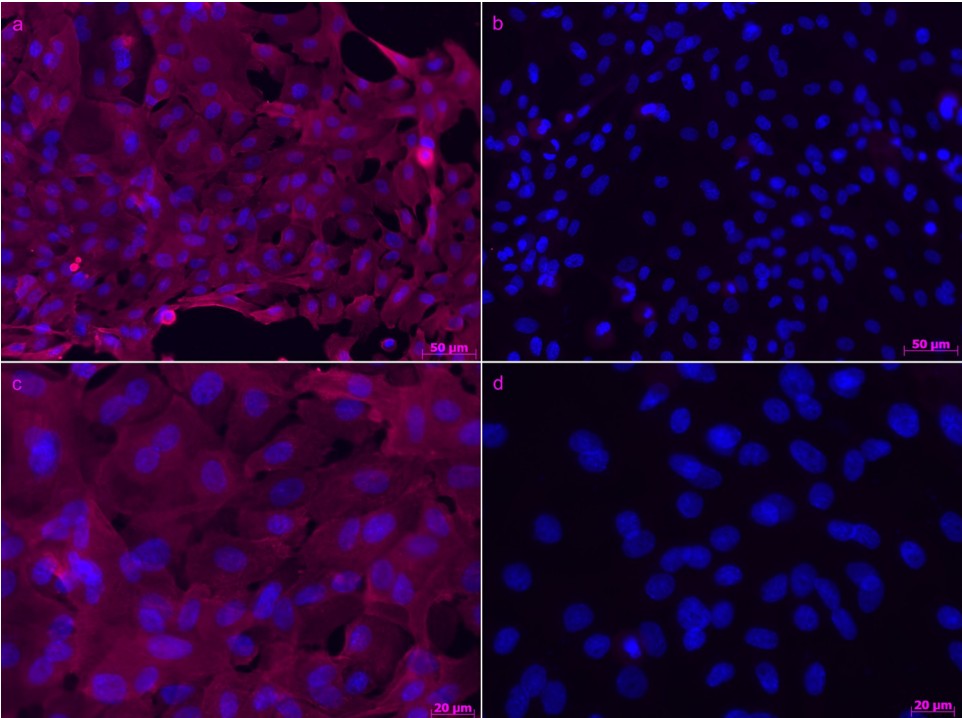

**Fig 1. Primary equine keratinocyte culture: Primary equine keratinocytes were stained with anti-cytokeratin.** (A) and (C) Staining of primary keratinocyte cultures with anti-cytokeratin: Cytoplasmic cytokeratin is shown in pink; nuclei were counterstained with Hoechst and are shown in blue; (A) 20x magnification; (C) 40x magnification. (B) and (D) Staining of keratinocytes with anti-vimentin: No staining was observed with this antibody. Nuclei are shown in blue; (B) 20x magnification; (D) 40x magnification.

minutes (Qiagen). Contaminating genomic DNA was removed by on-column DNase treatment, and RNA was quantified spectrophotometrically at 260 nm (NanoDrop 2000c, Thermo-Scientific, Reinach, Switzerland). Samples were subsequently handed to the Next Genome Sequencing platform of the University of Bern for RNA sequencing. RNA quality was determined using Fragment Bioanalyzer (Labgene, Châtel-Saint-Denis, Switzerland).

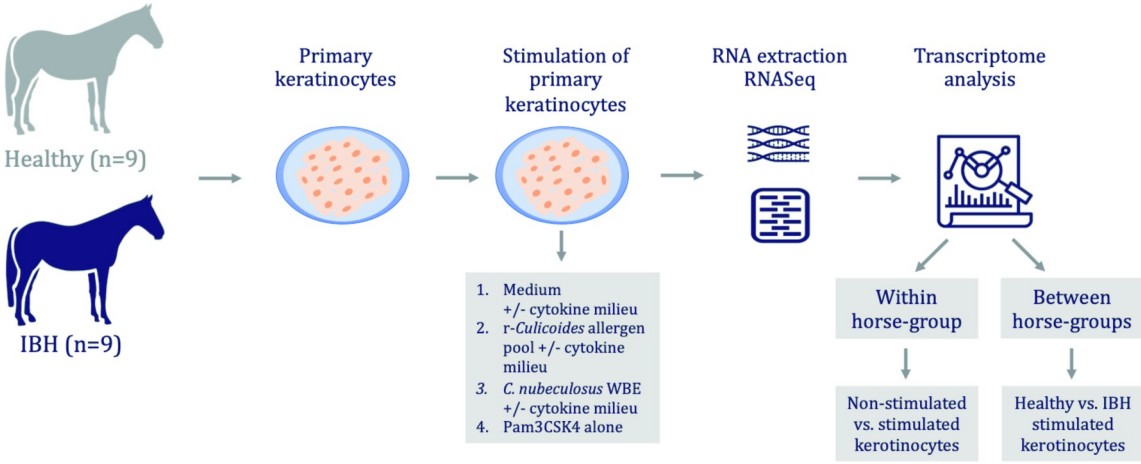

**Fig 2. Summary of the experimental setup.**

## RNA sequencing

Illumina TruSeq stranded mRNA libraries were prepared according to the manufacturer's protocol (Illumina, San Diego, USA). Between 17–31 mil 2 x 50 bp read-pairs per sample were generated on an Illumina NovaSeq 6000 instrument. The quality of the RNA-seq data was assessed using fastqc v. 0.11.5 and RSeQC v. 2.6.4.

## Mapping to reference genome and differential gene expression analysis

Differential gene expression analysis was performed as described in Cvitas et al. [34]. Briefly, reads were mapped to the reference genome (EquCab3.0) using HiSat2 v. 2.1.0 and Feature-Counts v. 1.6.0 was used to count the number of reads overlapping with each gene as specified in the genome annotation (NCBI *Equus caballus* Annotation Release 103). The Bioconductor package DESeq2 v. 1.18.1 was used for differential gene expression analysis.

To test for differential gene expression between the experimental groups we combined the factors "group" (IBH-affected or control) and "treatment" (unstimulated or six different keratinocyte stimulations) into a single factor with all combinations of the original factors (e.g. ibh_unstimulated for samples from unstimulated cells of IBH horses, h_unstimulated for a sample from unstimulated cells of healthy horses etc.) as described in the DESeq2-Vignette (http://bioconductor.org/packages/devel/bioc/vignettes/DESeq2/inst/doc/DESeq2.html#interactions). This resulted in a factor with 14 different combinations/levels (2 groups x 7 treatments) which we used to specify the comparisons i.e., contrast.

The Benjamini Hochberg method was used to correct for multiple testing. We did not remove any genes with low or no expression before running the DESeq analysis as the tool's "result" function performs an "independent filtering" by default which is based on the mean of normalized counts (see DESeq2 documentation on Bioconductor). Genes with a false discovery rate (= p adjusted) smaller than 0.05, and log2 fold change >1 were considered significantly differentially expressed. The datasets generated during the current study are available in the ENA repository via accession numbers PRJEB37568.

## Gene ontology analysis

TopGo v. 2.24.0 was used to identify gene ontology terms significantly enriched for differentially expressed genes (threshold for genes to be significantly differentially expressed: padjusted < 0.05). All tests were repeated using different combinations of algorithm (weight01 or classic) and test statistic (Fisher or Kolmogorov-Smirnov) to assess the robustness of the results. An interactive Shiny application was set up to facilitate the exploration and visualisation of the RNA-seq analysis results. All analyses were run in R version 3.4.4 (2018-03-15).

## Pathway analysis

ClusterProfiler v3.10.1 was used to test for enrichment of KEGG pathways with significantly differentially expressed genes. Gene set enrichment analysis (GSEA) was performed using the gseKEGG-function (default settings except for minGSSize = 50) and a ranked list as input (entrezgene-id and it's corresponding–log (raw pvalue), list sorted according to–log (raw pvalue).

# Results

Sequencing data was generated from all of the samples, with exception of samples of three IBH-KER and four H-KER cultures stimulated with Pam3CSK4, as these libraries did not produce enough sequencing reads. Therefore, we only performed the transcriptome analysis with

six IBH-KER and five H-KER Pam3CSK4-stimulated cultures. Data derived from one control horse was excluded from the analyses because it had a different expression profile than other horses, and we found expression of some genes that cells of the epithelial origin should not express, suggesting possible contamination of the sample.

## Stimulation of primary keratinocytes with *Culicoides* allergens did not induce changes in their gene expression

In order to investigate whether and how primary keratinocytes possibly contribute to pathogenesis of equine IBH, we stimulated keratinocytes with *C. nubeculosus* WBE and the pool of r-*Culicoides* allergens.

When comparing transcriptomes of IBH-KER and H-KER stimulated with WBE or r-*Culicoides* allergens to unstimulated IBH-KER and H-KER, we found no differentially expressed genes (DEGs; Fig 3A–3D). This was already noticeable in the results of the principal component analysis (PCA) based on 500 most variable genes, where samples of keratinocytes stimulated with *Culicoides* allergens clustered closely with non-stimulated keratinocytes (S1 Fig).

## Stimulation of primary keratinocytes with the allergic cytokine milieu is characterized by transcriptional changes in immune signatures and epithelial barrier

In order to understand how an allergic microenvironment might affect keratinocytes, we stimulated IBH-KER and H-KER with a combination of recombinant equine IL-4 and TNF-α, mimicking an allergic inflammatory milieu. When comparing IBH-KER stimulated with ACM to non-stimulated IBH-KER, we found 657 DEGs. Three hundred and seventeen (317) DEGs were significantly upregulated and 340 were significantly downregulated (Fig 3E). In H-KER, 413 significantly upregulated and 299 significantly downregulated DEGs were found (Fig 3F). Hierarchical clustering of non-stimulated samples and stimulated samples based on top 30 DEGs showed a clearly separated clustering of samples based on the culture conditions, in both IBH-KER and H-KER. 80% of the top 30 DEGs were shared between IBH-KER and H-KER. Genes involved in the inflammatory response (*NFKB1*, *ROR1*, *CXCL8*), cytokine mediated signaling (*IL31*, *IL23A*, *CISH*) as well as epithelial barrier formation (*KRT80*, *KRT7*) were among the top 30 DEGs (Fig 4A and 4B).

Gene ontology (GO) analysis of DEGs between non-stimulated and ACM stimulated IBH-KER indicated enrichment of processes such as inflammatory response, cytokine- and chemokine-mediated signaling and, interestingly, processes of keratinocyte differentiation, hair follicle development and regulation of hair follicle development (Tables 2 and S2).

Similarly, in H-KER biological processes involved in immune response such as inflammatory response, regulation of T cell chemotaxis, neutrophil chemotaxis as well as processes of hair follicle development and the regulation of the hair follicle development were enriched (Tables 3 and S3). Subsequently, we examined genes belonging to the enriched GO categories.

a) **Transcriptional changes of immune signature.** After stimulation with ACM, IBH-KER significantly upregulated expression of genes encoding various interleukins, such as *IL31*, *IL23A*, *IL36G*, *IL34*, *IL6*, and *IL1A*. Additionally, they also significantly upregulated expression of genes encoding different cytokines such as *CCL20*, *CCL27*, *CCL5* and *CCL11* as well as chemokines like *CXCL2*, *CXCL6*, *CXCL8*, *CXCL10* and *CXCL11*. Similarly, H-KER upregulated the same interleukins, and chemokines with exception of *CXCL10* and *CXCL11* (Table 4A). Interestingly, in IBH-KER the atopic cytokine milieu did not induce expression of epithelial-derived cytokines *TSLP*, *IL25* and *IL33*, known to play a major role in development of allergic

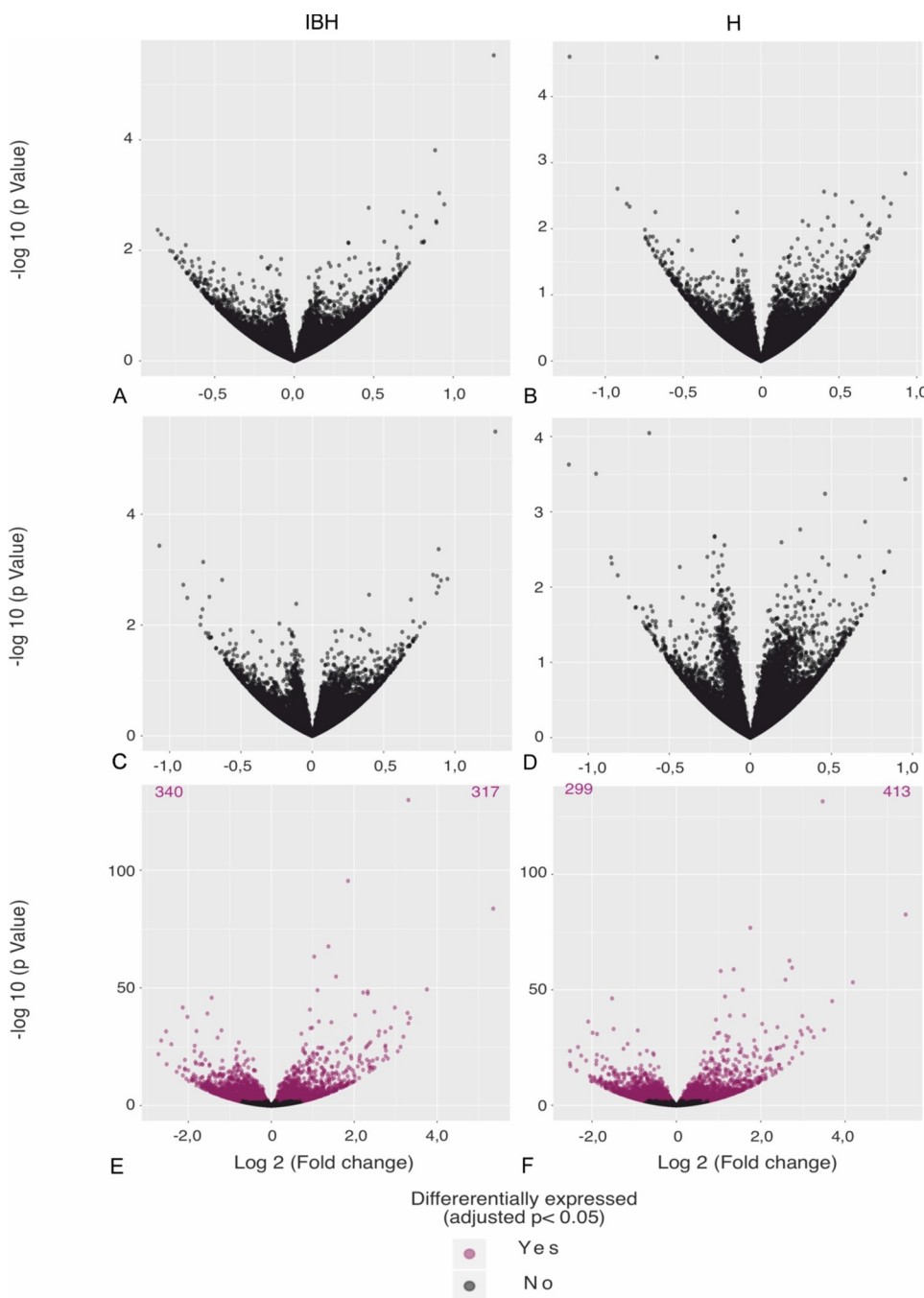

**Fig 3.** A-F. Volcano plots of significant DEGs in following comparisons: Non-stimulated (NS) vs. recombinant allergen pool stimulated, in IBH-KER (A) and H-KER (B). NS vs. *C. nubeculosus* WBE stimulated in IBH-KER (C) and H-KER (D). NS vs. ACM stimulated in IBH-KER (E) and H-KER (F).

inflammation (Table 4A). In H-KER, only *TSLP* was significantly upregulated following stimulation with the ACM (log2fold change 0.93) (Table 4A).

Gene set enrichment analysis (GSEA) using KEGG pathways additionally indicated significant overrepresentation of genes belonging to cytokine-cytokine receptor interaction,

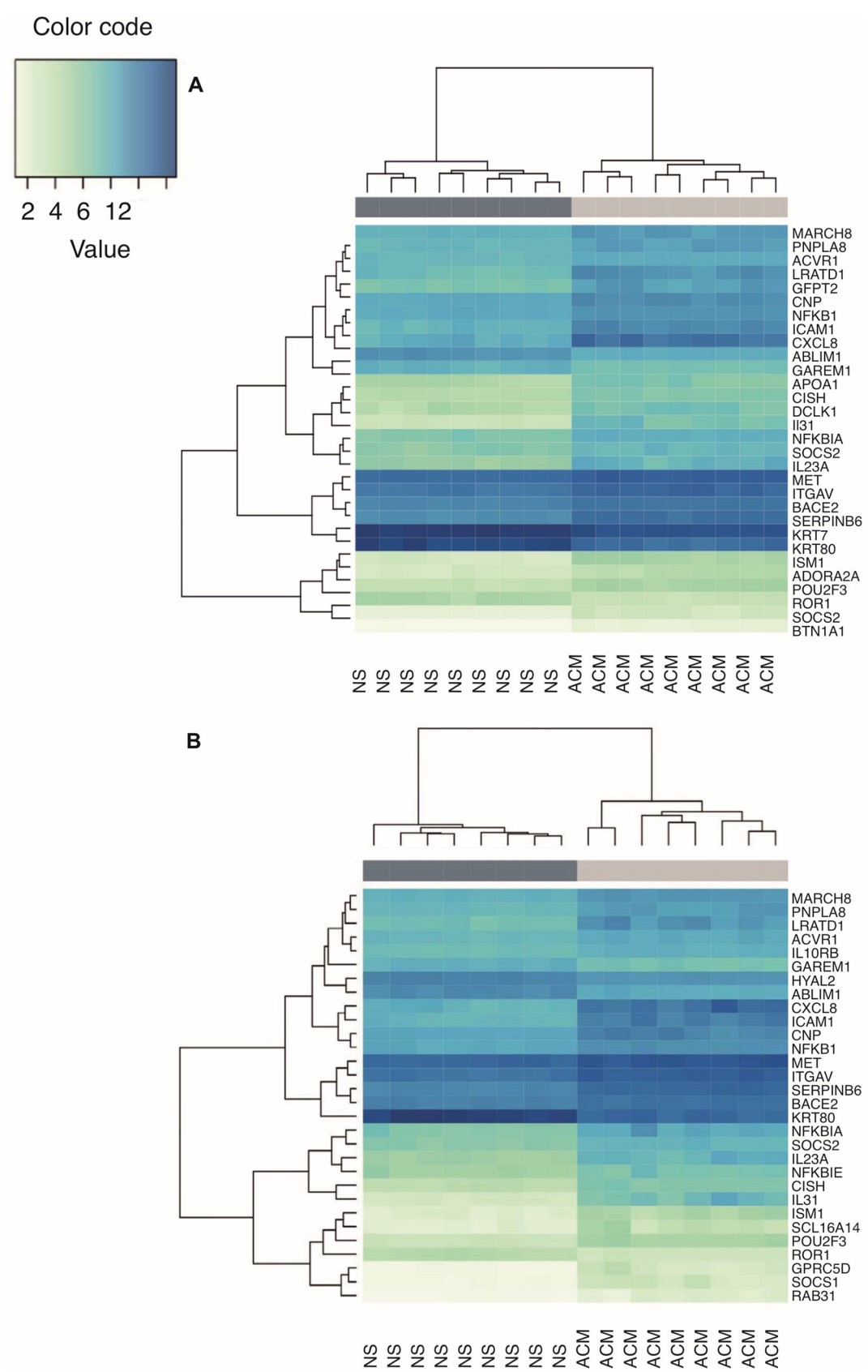

**Fig 4.** A-B Hierarchical clustering gene expression of top 30 genes of allergic cytokine milieu-stimulated and non-stimulated samples from the two compared conditions in both IBH-KER (A) and H-KER (B). Lower mean counts are shown in light green and higher mean counts in dark blue.

chemokine signaling, T cell and B cell receptor signaling pathways and Th1 and Th2 cell differentiation in both IBH-KER and H-KER (S4 and S5 Tables).

Interestingly, *IL31* was the highest upregulated gene when comparing ACM-stimulated and non-stimulated cells, in both IBH-KER and H-KER (log2fold change 5.35 and 5.44, respectively). Moreover, IL31 receptor subunit, *OSMR*, was significantly upregulated in both IBH-KER and H-KER (log2fold change 1.69 and 1.60). *IL31RA* subunit was significantly upregulated only in IBH-KER, however with lower log2 fold change (log2fold change. = 0.45) (Table 4A). Additionally, JAK-Stat signaling pathway through which IL-31 signals, was also significantly overrepresented in IBH-KER and H-KER stimulated with ACM (S4 and S5 Tables).

b) **Transcriptional changes of epithelial barrier.** GO analysis indicated that among the enriched biological processes, genes belonging to keratinocyte differentiation and hair follicle development were enriched in keratinocytes stimulated with allergic milieu (Tables 2 and 3). Therefore, we investigated the expression of genes belonging to these processes (Table 4B). Genes involved in keratinocyte differentiation such as *SCEL*, *KRT7*, *KRT13*, *KRT16*, and *KRT80* among others were significantly downregulated. *KRT6B* and *KRT14* were also significantly downregulated, however with lower log2fold change (-0.76 and -0.77, respectively). Only *KRT8* expression was upregulated in both IBH-KER and H-KER (log2 fold change = 0.89 and 1.02, respectively). The ACM did not influence the expression of major genes involved in terminal differentiation of keratinocytes like *FLG* and *IVL*. Furthermore, genes involved in homeostasis of epithelial lipids, such as *ALOXE3* and *ALOX12B* were significantly downregulated (log2 fold change -1.30 and -1.43, respectively).

In our previous work on transcriptome of lesional skin of IBH-affected horses, we reported significant downregulation of FGFR1 and ligands of FGFR2 in lesional skin of IBH-affected horses [34]. When we investigated the state of FGF receptors and ligands in IBH-KER, we found significant downregulation of *FGFR2* and *FGF9* as well as *FGF22* ligands following exposure to ACM. Expression of *FGFR1* was not affected. In H-KER only expression of *FGFR2* and *FGF9* was significantly downregulated (Table 4B).

Lastly, exposing keratinocytes to the pool of recombinant *Culicoides* allergens or WBE in combination with ACM resulted only in few significantly upregulated genes compared to stimulation with ACM only (S6, S7, S8 and S9 Tables), which were thus not further analyzed.

**Table 2. Selected biological processes enriched in IBH-KER stimulated with allergic cytokine milieu in comparison to non-stimulated IBH-KER.**

| GO-ID | Term | Annotated | Significant | Expected | Classic Fisher |
|---|---|---|---|---|---|
| GO:0007229 | Integrin-mediated signaling | 71 | 34 | 18.46 | 5.80E-05 |
| GO:0030593 | Neutrophil chemotaxis | 46 | 21 | 11.96 | 0.00042 |
| GO:0070098 | Chemokine-mediated signaling | 34 | 17 | 8.84 | 0.0012 |
| GO:0019885 | Antigen processing and presentation | 8 | 7 | 2.08 | 0.00043 |
| GO:0006954 | Inflammatory response | 298 | 109 | 77.48 | 3.00E-05 |
| GO:0019221 | Cytokine-mediated signaling | 252 | 109 | 65.52 | 1.50E-09 |
| GO:0010634 | Positive regulation of epithelial cell | 101 | 36 | 26.26 | 0.01996 |
| GO:0030216 | Keratinocyte differentiation | 69 | 33 | 17.94 | 7.60E-05 |
| GO:0051798 | Positive regulation of hair follicle development | 10 | 8 | 2.6 | 0.00055 |
| GO:0001942 | Hair follicle development | 67 | 33 | 17.42 | 3.60E-05 |

**Table 3. Selected biological processes enriched in H-KER stimulated with allergic cytokine milieu in comparison to non-stimulated H-KER.**

| GO.ID | Term | Annotated | Significant | Expected | Classic Fisher |
|-------|------|-----------|-------------|----------|----------------|
| GO:0001942 | Hair follicle development | 69 | 32 | 16.22 | 2.50E-05 |
| GO:0006954 | Inflammatory response | 304 | 117 | 71.45 | 2.40E-09 |
| GO:0019885 | Antigen processing and presentation | 8 | 7 | 1.88 | 0.00025 |
| GO:0051798 | Positive reg. of hair follicle development | 10 | 8 | 2.35 | 0.00026 |
| GO:0051092 | Positive reg. of NF-kappaB TF | 92 | 36 | 21.62 | 0.00057 |
| GO:0010820 | Positive reg. of T cell chemotaxis | 7 | 6 | 1.65 | 0.00094 |
| GO:0030593 | Neutrophil chemotaxis | 47 | 20 | 11.05 | 0.00294 |
| GO:0006955 | Immune response | 702 | 227 | 165 | 2.40E-08 |
| GO:0071347 | Cellular response to interleukin-1 | 51 | 25 | 11.99 | 0.000061 |
| GO:0022407 | Regulation of cell-cell adhesion | 250 | 97 | 58.76 | 3.50E-08 |

## Stimulation of primary IBH-KER and H-KER with TLR 1/2-ligand is characterized by transcriptional changes indicative of innate immune responses and impairment in cell proliferation

Lesions of IBH can be further exacerbated by secondary bacterial infections. In order to understand how keratinocytes respond to bacteria, we stimulated primary keratinocytes with the TLR 1/2 synthetic ligand, Pam3CSK4. This stimulation resulted in 206 significantly upregulated and 84 significantly downregulated DEGs in IBH-KER when compared to non-stimulated IBH-KER. In H-KER, 211 genes were significantly upregulated and 169 were significantly downregulated, when compared to non-stimulated H-KER (Fig 5A and 5B).

Non-stimulated and Pam3CSK4-stimulated keratinocytes clustered separately, both in case of IBH-KER and H-KER, as shown in Fig 6. Genes involved in regulation of inflammatory response (T*NFAIP3*, *TNF*), NF-κβ (*NFKBIZ*, *KFKB1*, *NFKBIA)* and chemokine signaling *(CXCL6*, *CXCL8)* were noticeable among the top 30 DEGs (Fig 6).

Interestingly, GO analysis showed that among the top 10 enriched biological processes in IBH-KER and H-KER stimulated with Pam3CSK4 were processes involved in cell cycle, i.e. cell division. Most of the DEGs belonging to processes of cell division were downregulated in both IBH-KER and H-KER (71.4% and 71.43%, respectively). 81% of DEGs belonging to a mitotic cell cycle process in IBH-KER were downregulated and 97.3% of DEGs belonging to the DNA replication process were downregulated, as well (5, 6, S10 and S11 Tables). Furthermore, Kegg pathway based GSEA showed that along with DNA replication and cell cycle pathways, pathways such as Toll like receptor signaling, NOD-like receptor signaling, C-type lectin signaling, Nf-κB signaling pathway were significantly overrepresented in IBH-KER and H-KER stimulated with Pam3CSK4 (S12 and S13 Tables).

When we examined genes belonging to these pathways, we found significant upregulation of *IL1A*, *IL23A*, *IL6* and *CSF2* and *CSF3* in IBH-KER. Chemokines such as *CCL20*, *CXCL1*, *CXCL2*, *CXCL6* and *CXCL8* were also significantly upregulated, suggesting strong innate immune activity of keratinocytes (Table 7). We also found significant upregulation of *TLR1*, *TLR6* and *TLR10*. Log2 fold change of these genes, was however, low (0.48–0.74, respectively). When we further investigated expression of genes belonging to NF-κβ signaling pathway, we found upregulation of *NFKB1* and *NFKB2*, however with lower log2 fold change (0.85 and 0.90) as well as *NFKBIA*, *NFKBI7* and *NFKBIE* (Table 7). Expression of most of these genes was similar in H-KER, with the exception of *IL6* and *TLR10*; their expression did not differ between stimulated and non-stimulated H-KER (Table 7).

**Table 4. DEGs are classified by gene families that influence (A) immune responses and (B) epithelial barrier formation and maintenance.** A-B. Cells were analyzed by RNA-sequencing and gene expression was compared between IBH-KER and H-KER stimulated with allergic cytokine milieu (ACM) or unstimulated keratinocytes derived from IBH-affected and H-horses. In **(A)** representative genes of immune responses and **(B)** epithelial barrier genes are shown. (Pink = statistically significant upregulation and log2 fold change >1; beige = statistically significant upregulation and log2 fold change <1; dark blue = statistically significant downregulation and log2 fold change >-1; light blue = statistically significant downregulation and log2 fold change <-1; gray = no difference in gene expression; false discovery rate <0.05). Log2 fold changes are noted for all DEGs.

**A**

| Functional group | Gene symbol | IBH-KER | H-KER |
|---|---|---|---|
| | | NS vs. ACM | NS vs. ACM |
| Immune signatures | IL31 | 5.35 | 5.44 |
| | IL23A | 2.98 | 2.99 |
| | CXCL8 | 2.88 | 2.95 |
| | CXCL10 | 2.81 | |
| | CXCL11 | 2.70 | |
| | IL36G | 2.35 | 2.38 |
| | CCL20 | 2.08 | 2.04 |
| | CCL11 | 1.35 | 1.98 |
| | CXCL6 | 1.52 | 1.91 |
| | IL6 | 1.37 | 1.78 |
| | OSMR | 1.69 | 1.60 |
| | IL34 | 1.27 | 1.59 |
| | CXCL1 | 1.22 | 1.52 |
| | CCL27 | 1.31 | 1.44 |
| | IL36RN | 1.21 | 1.23 |
| | CCL5 | | 1.20 |
| | IL1A | 1.04 | 1.07 |
| | TSLP | | 0.93 |
| | IL31RA | 0.45 | |
| | IL25 | | |
| | IL33 | | |

**B.**

| Functional group | Gene symbol | IBH-KER | H-KER |
|---|---|---|---|
| | | NS vs. ACM | NS vs. ACM |
| Epithelial barrier | KRT80 | -2.14 | -2.09 |
| | KRT16 | -1.85 | -1.75 |
| | SCEL | -1.46 | -1.41 |
| | ALOX12N | -1.45 | -1.47 |
| | ALOXE3 | -1.38 | -1.37 |
| | KRT13 | -1.35 | -1.46 |
| | KRT7 | -1.20 | -1.15 |
| | FGF9 | -1.14 | -1.24 |
| | KRT4 | -0.75 | -1.23 |
| | FGFR2 | -1.05 | -1.08 |
| | FGF22 | -1.03 | |
| | KRT6B | -0.76 | -0.80 |
| | KRT14 | -0.76 | -0.74 |
| | FGF1 | | |
| | KRT8 | 0.89 | 1.02 |
| | FGF2 | 1.09 | 1.08 |
| | FGFR1 | | |

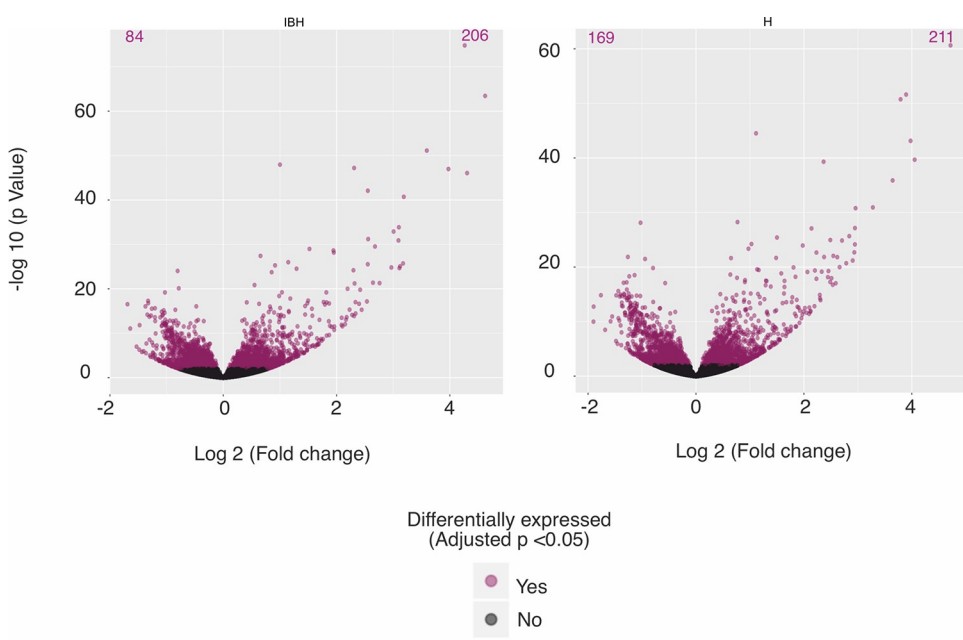

**Fig 5.** A-B. Volcano plots of significant DEGs in following comparisons: Non-stimulated (NS) vs.TLR 1/2 ligand (Pam3CSK4) of IBH-KER (A) and H-KER (B).

## Transcriptional differences between IBH-KER and H-KER

In order to investigate whether gene expression in keratinocytes derived from IBH-affected and control horses fundamentally differs, we first compared transcriptomes of non-stimulated keratinocytes derived from IBH-affected and control horses. We found expression of only one gene, *CTSL*, coding for cathepsin L1 to be significantly upregulated in IBH-KER compared to H-KER (S2 Fig). We furthermore wanted to investigate whether the response of primary equine keratinocytes derived from IBH-affected and control horses differs in response to the stimuli described above. Therefore, we investigated the differences in gene expression between IBH-KER and H-KER stimulated with ACM as well as the TLR 1/2 ligand. Because *Culicoides* allergen stimulation did not induce any significant changes in comparison to unstimulated keratinocytes (see above), differences in gene expression between IBH-KER and H-KER were not compared.

The transcriptomes of IBH-KER and H-KER stimulated with ACM differed in 23 DEGs (p < 0.05, log2 fold change > 1). Eighteen of those were significantly upregulated and five were significantly downregulated in IBH-KER compared to H-KER (S14 Table). Among upregulated DEGs in IBH-KER were, for example, *CXCL10*, *CXCL11 (p<0.0001, log2 fold change > 1.9)*, genes involved in chemokine signaling and genes such as *IFIH1*, *IFIH2*, *IFIT3* and *IFI44L*, encoding for proteins involved in interferon signaling. Genes such as *CH25H* and *IL34* were significantly downregulated in stimulated IBH-KER (S14 Table), however the FDR for these two genes was close to 5% and they may thus be artifacts.

Transcriptomes of IBH-KER and H-KER stimulated with Pam3CSK4 differed in seven DEGs (p < 0.05, log2fold change >1; S15 Table). Since only a low number of DEGs between the two study groups was found in both keratinocytes stimulated ACM or Pam3CSK4, GO and enrichment analyses were not performed.

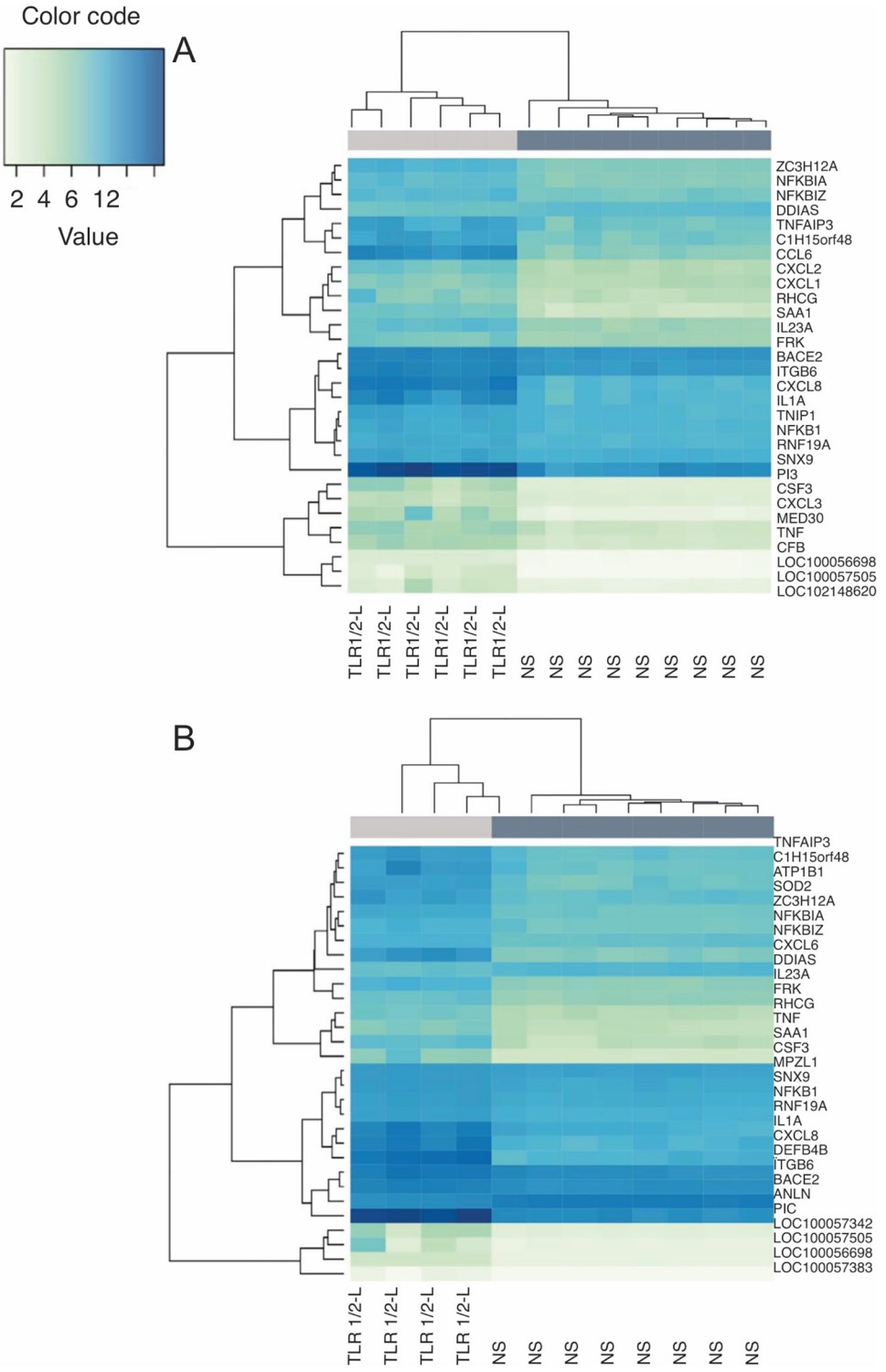

**Fig 6.** A-B. Hierarchical clustering gene expression of top 30 genes of TLR 1/2 ligand (Pam3CSK4)-stimulated and non-stimulated samples from following conditions: IBH-KER (A) and H-KER (B). Lower mean counts are shown in light green and higher mean counts in dark blue.

**Table 5. The 10 most significant biological processes enriched in IBH-KER stimulated with the TLR 1/2 synthetic ligand, Pam3CSK4, in comparison to non-stimulated IBH-KER.**

| GO-ID | Term | Annotated | Significant | Expected | Classic Fisher |
|---|---|---|---|---|---|
| GO:0000278 | Mitotic cell cycle | 656 | 231 | 137.78 | 2.90E-18 |
| GO:0007059 | Chromosome segregation | 254 | 106 | 53.35 | 3.30E-14 |
| GO:0000070 | Mitotic sister chromatid segregation | 127 | 62 | 26.67 | 2.40E-12 |
| GO:0006281 | DNA repair | 364 | 127 | 76.45 | 3.30E-10 |
| GO:0007093 | Mitotic cell cycle checkpoint | 100 | 47 | 21 | 5.20E-09 |
| GO:0007052 | Mitotic spindle organization | 90 | 40 | 18.9 | 4.60E-07 |
| GO:0007094 | Mitotic spindle assembly checkpoint | 25 | 16 | 5.25 | 4.00E-06 |
| GO:0000724 | Double strand break repair via homologous recombination | 114 | 45 | 23.94 | 5.00E-06 |
| GO:0006271 | DNA strand elongation | 16 | 12 | 3.36 | 5.60E-06 |
| GO:0051301 | Cell division | 211 | 70 | 44.32 | 2.20E-05 |

## Discussion

The role of epithelial barriers in the pathogenesis of Type I hypersensitivities is well-established in human allergy but is only poorly investigated in equine patients. Recently, we reported transcriptome data suggestive of alterations of the epithelial barrier in horses affected with insect bite hypersensitivity. We showed that lesional skin of IBH-affected horses is characterized by transcriptomic evidence of epithelial barrier disruption that is most likely immune mediated. We also found that non-lesional epidermis of IBH-affected horses shows transcriptomic evidence of lipid metabolism disruption and pruritus development which could act as predisposing factor for IBH [34]. In order to investigate a possible role of keratinocytes in the development of Type I hypersensitivities we studied transcriptomes of non-lesional IBH-KER and H-KER and their response to stimulation with *Culicoides* allergens, ACM and Toll like receptor 1/2 ligand (Pam3CSK4). Unstimulated keratinocytes from IBH-affected and H control horse were very similar at the transcriptional level, as seen by the absence of DEGs, except for *CTSL*, the gene coding for cathepsin L1. To investigate the response of equine keratinocytes to *Culicoides* allergens, we stimulated them with a pool of five r-*Culicoides* allergens as well as whole body extract of *C. nubeculosus*. The keratinocyte transcriptome did not change under the influence of either the r-allergens or WBE. *E.coli* expressed allergens are often not suitable for cellular assays [18], because they are often insoluble in inclusion bodies, have to be refolded and lack critical posttranslational modification. Additionally, endotoxin contamination can result in high background cytokine production. For these reasons we used five relevant

**Table 6. Top 10 Biological processes enriched in H-KER stimulated with the TLR 1/2 synthetic ligand, Pam3CSK4, in comparison to non-stimulated IBH-KER.**

| GO.ID | Term | Annotated | Significant | Expected | Classic Fisher |
|---|---|---|---|---|---|
| GO:0051301 | Cell division | 211 | 77 | 43.35 | 4.90E-08 |
| GO:0010950 | Positive regulation of endopeptidase activity | 106 | 29 | 21.78 | 0.05569 |
| GO:0034501 | Protein localization to kinetochore | 16 | 11 | 3.29 | 4.20E-05 |
| GO:0031297 | Replication fork processing | 32 | 17 | 6.57 | 4.60E-05 |
| GO:0006271 | DNA strand elongation involved in DNA replication | 16 | 12 | 3.29 | 4.40E-06 |
| GO:0000281 | Mitotic cytokinesis | 47 | 20 | 9.66 | 0.0005 |
| GO:0032922 | Circadian regulation of gene expression | 45 | 20 | 9.24 | 0.00025 |
| GO:0032508 | DNA duplex unwinding | 59 | 27 | 12.12 | 1.10E-05 |
| GO:0051988 | Regulation of attachment of spindle microtubules to kinetochore | 11 | 7 | 2.26 | 0.00231 |
| GO:0034080 | CENP-A containing nucleosome assembly | 5 | 5 | 1.03 | 0.00036 |

**Table 7. DEGs are classified by gene families that influence immune responses in the comparison of IBH-KER and H-KER stimulated with Pam3CSK4 and non-stimulated keratinocytes.** Cell samples were analyzed by RNA-sequencing and gene expressions were compared between Pam3CSK4 stimulated keratinocytes from IBH-affected and H-horses and non-stimulated keratinocytes. Only representative genes are shown. (Pink = statistically significant upregulation and log2 fold change >1; beige = statistically significant upregulation and log2 fold change <1; gray = no difference in gene expression; false discovery rate <0.05). Log2 fold changes are noted for all DEGs.

| Functional group | Gene symbol | IBH-KER | H-KER |
|---|---|---|---|
| | | NS vs. Pam3CSK4 | NS vs. PAm3CSK4 |
| Immune signatures | CSF3 | 4.63 | 4.71 |
| | CXCL6 | 4.26 | 3.79 |
| | CXCL2 | 3.19 | 2.33 |
| | CXCL8 | 3.10 | 2.95 |
| | IL23A | 2.68 | 2.96 |
| | IL1A | 2.56 | 2.96 |
| | TNF | 2.64 | 2.95 |
| | CSF2 | 2.54 | 2.31 |
| | CXCL1 | 2.55 | 2.46 |
| | CCL20 | 2.41 | 2.44 |
| | IL36G | 2.19 | 2.23 |
| | NFKBIA | 1.94 | 1.98 |
| | NFKBIZ | 1.52 | 1.48 |
| | NFKBIE | 1.22 | 1.19 |
| | IL6 | 1.00 | 0.96 |
| | NFKB2 | 0.90 | 0.94 |
| | NFKB1 | 0.85 | 0.97 |
| | TRL10 | 0.78 | |
| | TRL6 | 0.59 | 0.74 |
| | TLR1 | 0.48 | 0.57 |

*Culicoides* r-allergens produced in *P. pastoris*. Unfortunately, the allergens expressed in *P. pastoris* that were available for our study did not include proteases [4]. Recent studies have demonstrated the presence of a much larger number of allergens and proteins in *Culicoides* saliva, including proteases which are able to disrupt the epithelial barrier and thereby activate keratinocytes [11,16]. Furthermore, it is known that in human allergology, many major allergens are proteases [39–41]. Therefore, we also stimulated primary keratinocytes with WBE of *C. nubeculosus*. A limitation of our study is that its protease activity was not measured prior to stimulation of keratinocytes. This could account for the lack of stimulation of keratinocytes, which in turn resulted in no DEGs. The use of crude WBE has many limitations, nevertheless, *C. nubeculosus* WBE have been used with satisfying results for the re-stimulation of PBMCs as well as in basophil activation tests [42–44]. Finally, *in vivo*, keratinocytes may not only be stimulated by components in the *Culicoides* saliva but also by the mechanical damage to the skin induced by the bites of *Culicoides* [11]. Moreover, for studying the baseline response of keratinocytes to *Culicoides* allergens and to an allergic inflammatory milieu, the keratinocytes used in our study were derived from the non-lesional skin of both IBH-affected and control horses. We collected skin samples or biopsies from the inner thigh, where *Culicoides* midges do not usually bite. Unfortunately, we could not investigate whether keratinocytes derived from lesional skin respond differently, because in our hands, it was not possible to establish pure keratinocyte culture from lesional skin. Keratinocytes isolated from lesional skin sites might differ from keratinocytes derived from non-lesional sites due to mechanical damage induced by biting of the midges, which may in turn prime the keratinocytes towards a stronger

response to allergens due to the damage of epithelial barrier in those sites. Additionally, cells that detach and start proliferating in primary keratinocyte cultures are the basal, proliferative cells. As they proliferate, their differentiation state increases, and they soon stop their replication and die around passage five or six. Therefore, for our experiments we used keratinocytes of passage three that are not fully differentiated yet. This might have also influenced our results, as it was shown that mainly the fully differentiated keratinocytes produce epithelial-derived cytokines such as TSLP [27]. All these factors might account for the lack of transcriptomic differences between *Culicoides*-stimulated and non-stimulated keratinocytes.

On the other hand, when stimulated with a combination of recombinant equine IL-4 and TNF-α, mimicking an ACM, both IBH-KER and H-KER responded by changes in their transcriptome, suggesting that IBH is not associated with differing responses of IBH-KER or H-KER to an allergic milieu per se, but rather to presence or absence of a local Th2 microenvironment, caused by activation of immune cells such as Th2 lymphocytes, eosinophils and basophils in allergic individuals, which then, secondarily, activates keratinocytes. While an imbalance between the Th2 and T regulatory immune response has been described in IBH, it still remains unknown what are the initiating factors that skew the immune response towards a Th2 response in allergic horses [44,45]. Studies in human patients indicate that the microbiome may play an important role in the development of allergic conditions [46]. However, there is scarce information for the horse and so far no evidence of such effect in IBH [47]. Other factors such as genetic and environmental factors contribute to susceptibility to IBH [4,48]. The age at first exposure to *Culicoides* allergens also seems to play a crucial role for development of IBH later in life. Horses born in an environment free of *Culicoides* and exported as adults to *Culicoides*-rich environments have a much higher prevalence of IBH than horses of the same breed, exported at young age or born in a Culicoides-rich environment [4,48].

Expectedly, after stimulation with ACM keratinocytes upregulated many genes involved in immune responses, including many chemokines and interleukins (Table 4A). Particularly, stimulation with the allergic inflammatory milieu induced a strong upregulation of *CCL27*, the cutaneous T cell-attracting chemokine which is one of the main cytokines involved in atopic dermatitis (AD) pathogenesis [49,50]. Furthermore, cells also upregulated *CCL20* known to be produced in epidermis with impaired permeability (Table 4A). Moreover, *CCL20* is also upregulated in human keratinocytes under the influence of TNF-α [51]. Interestingly, due to its involvement in pruritus development, IL-31 has recently been shown to be a therapeutic target in treatment of IBH [52]. In IBH-lesional skin, we recently reported upregulation of both subunits of the IL-31 receptor, *IL31RA* and *OSMR*, however, the expression of the cytokine itself was not significantly upregulated. Interestingly, in the present study, upon stimulation of keratinocytes with the ACM, the top significantly upregulated gene in both IBH-KER and H-KER was *IL31* (log2 fold change 5.35 and 5.44, respectively). This is the first evidence that equine keratinocytes are capable of producing the Th2 cytokine *IL31*, following stimulation with allergic micromilieu, and not with TLR 1/2-L. This, however, needs to be further confirmed at the protein level. Unfortunately, antibodies specific for equine IL-31 are not (yet) available. Additionally, IL-31 can also modify the formation of the skin barrier in multiple ways, as demonstrated in human patients. It downregulates the expression of filaggrin, known to be the major protein involved in terminal differentiation of human keratinocytes, weakens the lipid envelope formation and represses enzymes and proteins involved in desmosome formation [53,54]. The role of filaggrin in the pathogenesis of equine IBH has not been studied extensively, but in a transcriptomic study, there was no evidence of an altered expression of filaggrin in lesional IBH skin [55]. In human patients it has now been proposed that IL-31 is a key player in the pathogenesis of AD, and based on our data, IL-31 seems to play an important role

in the pathogenesis of IBH [56]. Indeed, upon stimulation with ACM, many genes involved in formation of epithelial barrier were significantly downregulated. *SCEL*, the gene encoding for sciellin, involved in terminal differentiation of keratinocytes, as well as *KRT16*, *KRT6B* and other types of keratin were significantly downregulated, suggesting immune-mediated disruption of the barrier. Interestingly, in our previous study, we found a significant downregulation of *SCEL* in lesional whole skin of IBH-horses as well as in non-lesional epidermis of IBH-affected horses [34]. Our data thus confirms the importance of sciellin in the epithelial barrier of horses. Furthermore, we found downregulation of *FGFR2* and its ligand *FGF9* as well as downregulation of *FGF1* and *FGF22*, both ligands of FGFR1. Yang *et al.* have described a fibroblast growth factor receptor 1 and 2 (*fgfr1*, *fgfr2*) knock out mouse model that develops skin lesions similar to those in patients with AD, particularly with regard to the inflammatory infiltrate and the epidermal thickening [57]. Yang *et al.* attributed the hyperproliferative phenotype to action of IL36B and the S100A8/S100A9 complex. We recently also found transcriptomic evidence of an impairment in FGFR signaling and tight junction disruptions in lesional skin of IBH horses, suggesting that this pathway may indeed play an important role in disruption of epithelial barrier in IBH-affected horses [34]. However, the exact mechanism remains to be elucidated. Taken together, an ACM-induced downregulation of genes involved in epithelial barrier formation suggests the disruption of epithelial barrier by an allergic microenvironment in the horse, similar to human patients [58–60].

IBH-lesional skin is characterized by a strong infiltration with eosinophils [4,61]. However, the mechanism of eosinophil influx still remains largely unknown. In our study, when keratinocytes were stimulated with the ACM, they significantly upregulated the expression of *CCL11* in both IBH-KER and H-KER. It has been shown that in human patients IL-4 can induce the production of CCL11 by keratinocytes [62]. Importantly, *ICAM1* gene coding for intracellular adhesion molecule 1 which plays a key role in adhesion of eosinophils was significantly upregulated in stimulated equine keratinocytes. *CCL5* was significantly upregulated in stimulated H-KER, suggesting a micromilieu-dependent eosinophil homing mechanism.

IBH lesions can be further exacerbated by secondary bacterial infections. In order to investigate how IBH-KER and H-KER respond to bacterial PAMPs, we also stimulated keratinocytes with the toll like receptor 1/2 ligand, Pam3CSK4. In response to this TLR-ligand, both IBH-KER and H-KER showed a strong response inducing TLR and its downstream MyD88 and NF-κβ signaling. Not surprisingly, expression of inflammatory genes was significantly upregulated. Interestingly, genes involved in formation of the epithelial barrier were not affected, unlike following stimulation with the ACM, suggesting that this type of response is specific to the allergic milieu.

Analysis of transcriptional difference between IBH-KER and H-KER stimulated with ACM yielded 23 DEGs and in keratinocytes stimulated with Pam3CSK4 only seven DEGs. However, considering that in our analysis FDR of 5% was taken into account, some of these genes may be artifacts. Furthermore, one healthy horse seems to be an outlier and reacting differently to stimulations (S1 Fig), which also accounts for the difference we saw when comparing IBH-KER and H-KER. This suggests that in this experimental setup, there is no clear transcriptomic difference between IBH-KER and H-KER.

Taken together, our data suggests that equine keratinocytes are, in fact, capable of responding to different stimuli and may play a role in the pathogenesis of IBH, acting as amplifiers of allergic immune reaction through their response to ACM, and thus contributing to the local skin damage in immune-mediator-dependent way. Stimulation with a limited panel of *Culicoides* r-allergens did not induce a response of keratinocyte. Further studies are needed to assess whether a disruption of the epidermal barrier through mechanical and/or protease

induced damage by *Culicoides* contributes to the initiations of the allergic immune response in IBH or whether skin dendritic cells, innate immune cells and T-cells are the major players.

## Supporting information

**S1 Fig. PCA analysis.**
(TIF)

**S2 Fig. Volcano plot of significant DEG in the comparison of non-stimulated IBH-KER and H-KER.**
(TIF)

**S1 Table. Experimental conditions.**
(XLSX)

**S2 Table. Full GO analysis of IBH-KER stimulated with allergic cytokine milieu.**
(XLSX)

**S3 Table. Full GO analysis of H-KER stimulated with allergic cytokine milieu.**
(XLSX)

**S4 Table. Full KEGG analysis of IBH-KER stimulated with allergic cytokine milieu.**
(XLSX)

**S5 Table. Full KEGG analysis of H-KER stimulated with allergic cytokine milieu.**
(XLSX)

**S6 Table. DEGs from IBH-KER stimulated with allergic cytokine milieu versus IBH-KER stimulated with allergic cytokine milieu in combination with *Culicoides* recombinant proteins.**
(XLSX)

**S7 Table. DEGs from H-KER stimulated with allergic cytokine milieu versus H-KER stimulated with allergic cytokine milieu in combination with *Culicoides* recombinant proteins.**
(XLSX)

**S8 Table. DEGs from H-KER stimulated with allergic cytokine milieu versus H-KER stimulated with allergic cytokine milieu in combination with *Culicoides* whole body extract.**
(XLSX)

**S9 Table. DEGs from IBH-KER stimulated with allergic cytokine milieu versus IBH-KER stimulated with allergic cytokine milieu in combination with *Culicoides* whole body extract.**
(XLSX)

**S10 Table. Full GO analysis of IBH-KER stim with Pam3CSK4.**
(XLSX)

**S11 Table. Full GO analysis of H-KER stim with Pam3CSK4.**
(XLSX)

**S12 Table. Full KEGG analysis of IBH-KER stim with Pam3CSK4.**
(XLSX)

**S13 Table. Full KEGG analysis of H-KER stim with Pam3CSK4.**
(XLSX)

**S14 Table. DEGs from IBH-KER + allergic cytokine milieu versus H-KER + allergic cytokine milieu.**
(XLSX)

**S15 Table. DE Gs from IBH-KER + Pam3CSK4 versus H-KER + Pam3CSK4.**
(XLSX)

## Acknowledgments

We thank Jelena Mirkovitch and Shui Chu Ling for their expert laboratory assistance. We are also thankful to the Next Generation Sequencing Platform of the University of Bern for performing the high-throughput sequencing experiments. We are grateful to Boehringer-Ingelheim Vetmedica GmbH, Dr. Dania Reiche, for kindly providing the recombinant *Culicoides* allergens, and to Professor Michael Stoffel, for his support with immunofluorescence staining. We thank Dr. Katharina Windbichler, Institute of Veterinary Anatomy, University of Bern, for making the immunofluorescence pictures. Microscopy was performed on equipment supported by the Microscopy Imaging Center (MIC), University of Bern, Switzerland. We also thank Dr Dania Reiche, Dr Katharina Windbichler and Professor Michael Stoffel for carefully reading the manuscript.

## Author Contributions

**Conceptualization:** Simone Oberhaensli, Tosso Leeb, Eliane Marti.

**Data curation:** Simone Oberhaensli.

**Formal analysis:** Iva Cvitas, Simone Oberhaensli.

**Funding acquisition:** Eliane Marti.

**Investigation:** Iva Cvitas.

**Methodology:** Iva Cvitas, Tosso Leeb, Eliane Marti.

**Project administration:** Iva Cvitas, Eliane Marti.

**Resources:** Eliane Marti.

**Supervision:** Eliane Marti.

**Validation:** Iva Cvitas.

**Visualization:** Iva Cvitas, Simone Oberhaensli.

**Writing – original draft:** Iva Cvitas.

**Writing – review & editing:** Simone Oberhaensli, Tosso Leeb, Eliane Marti.

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
