## [Decision Letter · Decision Letter 0]

31 Aug 2021

PONE-D-21-13837

Equine keratinocytes in the pathogenesis of insect bite hypersensitivity: just another brick in the wall?

PLOS ONE

Dear Dr. Marti,

Thank you for submitting your manuscript to PLOS ONE. After careful consideration, we feel that it has merit but does not fully meet PLOS ONE’s publication criteria as it currently stands. Therefore, we invite you to submit a revised version of the manuscript that addresses the points raised during the review process.

Both reviewers have raised some concerns regarding the manuscript that would need to be fully addressed carefully.

We look forward to receiving your revised manuscript.

Kind regards,

Angel Abuelo, DVM, MRes, MSc, PhD, DABVP (Dairy), DECBHM

Academic Editor

PLOS ONE

1. Please ensure that your manuscript meets PLOS ONE's style requirements, including those for file naming. The PLOS ONE style templates can be found at https://journals.plos.org/plosone/s/file?id=wjVg/PLOSOne_formatting_sample_main_body.pdf and https://journals.plos.org/plosone/s/file?id=ba62/PLOSOne_formatting_sample_title_authors_affiliations.pdf.

2. In your Methods section, please provide additional details regarding participant consent from the owners of the animals. In the ethics statement in the Methods and online submission information, please ensure that you have specified (1) whether consent was informed and (2) what type you obtained (for instance, written or verbal). If the need for consent was waived by the ethics committee, please include this information.

“We thank Jelena Mirkovitch and Shui Chu Ling for their expert laboratory assistance. Arthropods used in this study were provided by the Pirbright Institute under UK under grant code BBS/E/I/00007039 awarded to Dr Simon Carpenter as part of funding received from the Biotechnology and Biological Science Research Council (UKRI).”

“This work was supported by the Swiss

National Science Foundation grant no. 310030-160196/1. This SNF grant was awarded to E.M. http://www.snf.ch/

The funders had no role in study design, data collection and analysis, decision

to publish, or preparation of the manuscript”

6. We note that you have referenced (five r-allergens that had proven to induce sulfidoleukotriene release in IBH-affected but not in healthy horses in a basophil activation test) which has currently not yet been accepted for publication. Please remove this from your References and amend this to state in the body of your manuscript [Unpublished]”) as detailed online in our guide for authors

Additional Editor Comments (if provided):

Reviewers' comments:

Reviewer's Responses to Questions

**Comments to the Author**

1. Is the manuscript technically sound, and do the data support the conclusions?

Reviewer #1: Partly

Reviewer #2: Yes

2. Has the statistical analysis been performed appropriately and rigorously? 

Reviewer #1: Yes

Reviewer #2: Yes

3. Have the authors made all data underlying the findings in their manuscript fully available?

Reviewer #1: Yes

Reviewer #2: Yes

4. Is the manuscript presented in an intelligible fashion and written in standard English?

Reviewer #1: Yes

Reviewer #2: Yes

5. Review Comments to the Author

Reviewer #1: The papers describes the transcriptome of keratinocytes isolated from IBH- or healthy horses, non-lesional skin. The investigations and descriptions have been performed in depth, and presented in a clear manner.

The limitations of the study (no parts from lesional skin, missing recombinant allergens like proteases, no enzymatic activity testing of used WBE etc.) have been listed and discussed.

Major point:

* due to missing info for lesional skin and important allergens from Cul. saliva, and also in light of no differences between IBH and healthy horses as well as in practice not being able perform a longitudinal study (horses before and after IBH-development), I would tune down all the statements suggesting kerationcytes as being responsible in IBH development (e.g. line 25)!

* include in discussion the contribution of genetics/breed, environment, feed, hygiene treatment etc. of animals in IBH development

* as an ACM alters transcriptome in both, IBH and healthy horses, discuss a) where that ACM might come from in a natural setting and what might be different between allergy-susceptible and healthy animals; and b) if both IBH and healthy keratinocytes react to ACM, why don't healthy develop allergies (i.e. don't they ever have a situation/setting where ACM is around, or do they have an efficient regulatory response?).

Minors:

* short title should be short

* line 37: introduce abbreviations of IHB-KER/H-KER in line 26

* line 38: name the single gene that differed

* line 58: "...due to strong pruritus and inflicted self-trauma"

* line 72: "...have been shown..."; add why E.coli expressed proteins are of limited use

* line 76: complete with "The binding of IgE to the FceRI and crosslinking of bound IgE with allergen activates..."

* line 96: instead of natural model better use "a valuable source of information from horses with spontaneously ocuring disease for the role of kerationcytes in allergic skin diseases also for other species."

* line 99: include reference

* line 108: Pam3CSK4: add that it is a synthetic ligand

* Methods:

- include time-point of skin sampling and medium as well as temperature for transport and storage after slaughtering;

- include stimulation conditions (conc. of stimulations, incubation time, temperature etc.); Table S1 is not sufficient

- line 145: which staining?

- line 163: 70 or 80% confluence (stated at different places in MS)

- throughout MS, take care to put space between "pvalues"

- table 4 and 5 need different titles/sub-titles

* line 411: state in paragraph again that Cul. allergens did induce no changes, therefore were not compared between KER and H horses

* line 422: 23 DEGs were later discussed as artefacts - insert that info already in results section!

* line 453: "either" "or" instead of neither/nor

* 454: "had been proven"

* line 457: again explain why E.coli expr. proteins are not suitable

* line 459: insert "relevant" allergens

* lines 461-463: change order of sentences

* line 488: mimicking an allergic cytokine milieu with IL-4 and TNF-alpha: discuss again where that would come from in IBH horses vs. healthy

* line 500: "interestingly, in the present study upon stimulation..."

* line 515: should be sciellin

* Figures: please put in addition to figure legend the labels of the different figure parts also directly in figure for better legibility, e.g. left panels title IBH-KER vs. right panel titles IBH-H, and rec. allergens vs. WBE vs. ACM in Fig.2; like done in Fig. 4

* Discussion: is Filaggrin known to be important in horses as well (e.g. null-mutations)?

Reviewer #2: Dear Authors,

The submitted manuscript describes a role for equine keratinocytes in the mechanism of allergy (specifically equine IBH) by acting as bystander responders to allergy-associated cytokines. This provides a link between immune cells and epidermal cells in amplifying an allergic response. Please see below my review comments on your manuscript.

MAJOR COMMENTS

1. Please add more background on the selection of IL-4 and TNF-a, but not others, and what is known about these two cytokines in IBH pathogenesis. Why were other Th2 cytokines, such as IL-5 and IL-13, not included? How did you determine the concentrations of the ACM? How did you determine the concentrations of the r-allergen, WBE and TLR ligand stimulation conditions?

2. Please clarify the stimulation conditions in the methods. Line 165 should say “…WBE OR toll like…”. Figure 1 suggests that some samples were co-stimulated with r-allergens or WBE and also with ACM. However, those data are not presented. Either show the data from co-stimulations or clarify Figure 1.

3. Does pre-exposing KER to r-allergens/WBE change the transcriptional response after subsequent ACM stimulation? In other words, is there an additive effect of allergen and cytokine on the KER response?

4. Add a figure validating the purity and identity of keratinocyte cultures.

5. Figures 4 and 7 are challenging to interpret with many different color codes. I suggest adding FDR/p(adj) values as well. And/or change the order of genes in descending Log2 fold change so each color-coded category is grouped together. Graphical representation of these data would also be useful.

6. Please comment in the discussion how keratinocytes in lesional skin of IBH horses may differ from the keratinocyte samples compared in this study.

MINOR COMMENTS

1. Should line 187 say 17-31 MIL 2x50bp? Correct typo if it is one.

2. The figure titles and captions need to be improved. The first sentence should summarize the point of the figure (ex: in Fig 2 this could just be “Volcano plots of significant DEGs in different comparisons”), and the following unbolded sentences should describe the different subparts of the figure, describing sample types, color coding, etc. Currently, the figure titles are what should be the caption.

3. In Figures 2+5, it would be beneficial to label the genes of interest that are highlighted in Figures 4+7.

4. Line 452-453 is a double negative. Should be “did not change…either of the r-allergens….”

5. Line 460: activate

6. Line 556: artifacts

7. Line 556: outlier

6. PLOS authors have the option to publish the peer review history of their article (what does this mean?). If published, this will include your full peer review and any attached files.

Reviewer #1: No

Reviewer #2: No

---

## [Author Response · Author response to Decision Letter 0]

4 Feb 2022

Dear Editor, dear Reviewers,

We thank the reviewers and editor for their useful comments. We have corrected the manuscript accordingly. Below please find our response to the reviewer with the corresponding line numbers in the revised manuscript marked in red. 

1. Please ensure that your manuscript meets PLOS ONE's style requirements, including those for file naming. The PLOS ONE style templates can be found at https://journals.plos.org/plosone/s/file?id=wjVg/PLOSOne_formatting_sample_main_body.pdf and https://journals.plos.org/plosone/s/file?id=ba62/PLOSOne_formatting_sample_title_authors_affiliations.pdf.

Thank you for the comment, we think that the manuscript meets PLOS ONE's style requirements.

2. In your Methods section, please provide additional details regarding participant consent from the owners of the animals. In the ethics statement in the Methods and online submission information, 

please ensure that you have specified (1) whether consent was informed and (2) what type you obtained (for instance, written or verbal). If the need for consent was waived by the ethics committee, please include this information.

The samples were all taken from slaughter horses, except for one horse, for which written consent was obtained. This information is now provided in lines 138-139. 

check

“We thank Jelena Mirkovitch and Shui Chu Ling for their expert laboratory assistance. Arthropods used in this study were provided by the Pirbright Institute under UK under grant code BBS/E/I/00007039 awarded to Dr Simon Carpenter as part of funding received from the Biotechnology and Biological Science Research Council (UKRI).”

The Funding information has been removed from the acknowledgments.

Please update the Funding Statement as completed in red below:

“This work was supported by the Swiss National Science Foundation grant no. 310030-160196/1. This SNF grant was awarded to E.M. http://www.snf.ch/. Arthropods used in this study were provided by the Pirbright Institute under UK under grant code BBS/E/I/00007039 awarded to Dr Simon Carpenter as part of funding received from the Biotechnology and Biological Science Research Council (UKRI).

The funders had no role in study design, data collection and analysis, decision to publish, or preparation of the manuscript”

Repository information is provided in lines 240 to 241: “The datasets generated during the current study are available in the ENA repository via accession numbers PRJEB37568 “.

6. We note that you have referenced (five r-allergens that had proven to induce sulfidoleukotriene release in IBH-affected but not in healthy horses in a basophil activation test) which has currently not yet been accepted for publication. Please remove this from your References and amend this to state in the body of your manuscript [Unpublished]”) as detailed online in our guide for authors data as supplementary material 

We have removed this sentence from the text as we cannot provide the unpublished data. We would like to include this data in a later manuscript specific for the study that we had mentioned. 

We have one new co-author, who did the new figure 1.

 

5. Review Comments to the Author

Reviewer #1: The papers describes the transcriptome of keratinocytes isolated from IBH- or healthy horses, non-lesional skin. The investigations and descriptions have been performed in depth, and presented in a clear manner.

The limitations of the study (no parts from lesional skin, missing recombinant allergens like proteases, no enzymatic activity testing of used WBE etc.) have been listed and discussed.

Major point:

* due to missing info for lesional skin and important allergens from Cul. saliva, and also in light of no differences between IBH and healthy horses as well as in practice not being able perform a longitudinal study (horses before and after IBH-development), I would tune down all the statements suggesting kerationcytes as being responsible in IBH development (e.g. line 25)!

Thank you for your comment. 

Line 43: We have written that equine keratinocytes have the ability of eliciting differing responses to different stimuli and may possibly play a role in the pathogenesis of IBH. In the next lanes we explain that they are not the primary responders but that they can amplify the allergic immune response by their activation mediated by allergic inflammatory cytokines, such as IL-4 and TFN- a. 

Furthermore, in the discussion, lines 621-628 we address this matter and further elaborate it. 

* include in discussion the contribution of genetics/breed, environment, feed, hygiene treatment etc. of animals in IBH development

Thank you for the comment. This is now mentioned in lines 536-544.

* as an ACM alters transcriptome in both, IBH and healthy horses, discuss a) where that ACM might come from in a natural setting and what might be different between allergy-susceptible and healthy animals; and b) if both IBH and healthy keratinocytes react to ACM, why don't healthy develop allergies (i.e. don't they ever have a situation/setting where ACM is around, or do they have an efficient regulatory response?) ; 

Thank you for the insight. This is now added in the following lines: 92-97 and 529-536.

Minors: 

Thank you for the minor revisions. They are corrected as follows: 

* short title should be short : Corrected in line 17

* line 37: introduce abbreviations of IHB-KER/H-KER in line 26; line 27

* line 38: name the single gene that differed ; line 39

* line 58: "...due to strong pruritus and inflicted self-trauma" ; line 59

* line 72: "...have been shown..."; add why E.coli expressed proteins are of limited use ; 

This is now clarified in lines 72-74: … such as endotoxin contamination and lack of post-translational modification of the protein, leading to unspecific stimulations or lack of response, respectively, and thus limiting their use for cellular in vitro assays [19]. 

* line 76: complete with "The binding of IgE to the FceRI and crosslinking of bound IgE with allergen activates..." ;added to line 79

* line 96: instead of natural model better use "a valuable source of information from horses with spontaneously ocuring disease for the role of kerationcytes in allergic skin diseases also for other species." ; line 106-108

* line 99: include reference, line 110

* line 108: Pam3CSK4: add that it is a synthetic ligand ; line 118

* Methods:

- include time-point of skin sampling and medium as well as temperature for transport and storage after slaughtering; line 136-139 

- include stimulation conditions (conc. of stimulations, incubation time, temperature etc.); Table S1 is not sufficient; Table S1 edited

- line 145: which staining?; line 159

- line 163: 70 or 80% confluence (stated at different places in MS); lines 152, 158

- throughout MS, take care to put space between "pvalues"; corrected throughout the manuscript 

- table 4 and 5 need different titles/sub-titles

Thank you! The titles of table 4 and 5 have been corrected

* line 411: state in paragraph again that Cul. allergens did induce no changes, therefore were not compared between KER and H horses; 

This information is now added in lines 453-455.

* line 422: 23 DEGs were later discussed as artefacts. This was not well formulated and has been adapted (line 615) - insert that info already in results section!; has been added: lines 463-464

* line 453: "either" "or" instead of neither/nor ; line 489

* 454: "had been proven" ; rewritten in line 491

* line 457: again explain why E.coli expr. proteins are not suitable ; line 490-493

* line 459: insert "relevant" allergens ; line 493

* lines 461-463: change order of sentences ; line 495-498

* line 488: mimicking an allergic cytokine milieu with IL-4 and TNF-alpha: discuss again where that would come from in IBH horses vs. healthy ; line 531-533

* line 500: "interestingly, in the present study upon stimulation..." line 556

* line 515: should be sciellin; line 574

* Figures: please put in addition to figure legend the labels of the different figure parts also directly in figure for better legibility, e.g. left panels title IBH-KER vs. right panel titles IBH-H, and rec. allergens vs. WBE vs. ACM in Fig.2; like done in Fig. 4; 

This was now corrected in figures 3A-F, 4A-B, 6 A-B, 7 A-B 

* Discussion: is Filaggrin known to be important in horses as well (e.g. null-mutations)?; no there is no evidence so far, line 567-569

Reviewer #2: Dear Authors,

The submitted manuscript describes a role for equine keratinocytes in the mechanism of allergy (specifically equine IBH) by acting as bystander responders to allergy-associated cytokines. This provides a link between immune cells and epidermal cells in amplifying an allergic response. Please see below my review comments on your manuscript.

MAJOR COMMENTS

1. Please add more background on the selection of IL-4 and TNF-a, but not others, and what is known about these two cytokines in IBH pathogenesis. Why were other Th2 cytokines, such as IL-5 and IL-13, not included? How did you determine the concentrations of the ACM? 

How did you determine the concentrations of the r-allergen, WBE and TLR ligand stimulation conditions?;

Thank you for your comment. This is added in lines 92-97 and 194-196. At the time of the experimental part of the study, equine recombinant IL-5 and IL-13 were not, unlike nowadays, available. Additionally, we wanted to focus more on IL-4, being shown to be upregulated in the skin of IBH-affected horses. However, considering the novel data from human medicine showing that IL-13 seems to be the key player in the peripheral sights (e.g. skin) of atopic individuals, and our data from sequencing full lesional skin of IBH-horses that showed upregulation of IL13, it would have been very interesting to have a group of keratinocytes stimulated with IL-13 alone or in combination with IL-4 and TNF-alpha. 

2. Please clarify the stimulation conditions in the methods. Line 165 should say “…WBE OR toll like…”.

Thank you has been modified. Line 189

 Figure 1 suggests that some samples were co-stimulated with r-allergens or WBE and also with ACM. However, those data are not presented. Either show the data from co-stimulations or clarify Figure 1. ; 

Thank you for the valuable insight. Four supplementary tables (Table S6, -S7, -S8 and -S9) were added, and text in lines 376-379 was accordingly added. 

3. Does pre-exposing KER to r-allergens/WBE change the transcriptional response after subsequent ACM stimulation? In other words, is there an additive effect of allergen and cytokine on the KER response? 

Thank you for the comment. This is now added in lines 376-379.

4. Add a figure validating the purity and identity of keratinocyte cultures. 

Thank you for comment. A Figure was added as Figure 1. 

5. Figures 4 and 7 are challenging to interpret with many different color codes. I suggest adding FDR/p(adj) values as well. And/or change the order of genes in descending Log2 fold change so each color-coded category is grouped together. Graphical representation of these data would also be useful.

Thank you for the comment. Figures were changed so that change of the order of genes is in descending Log2 fold change and therefore each color-coded category is grouped together (now Figure 5 A-B and Figure 8). 

6. Please comment in the discussion how keratinocytes in lesional skin of IBH horses may differ from the keratinocyte samples compared in this study. 

Thank you for your insight. This is added in lines 514-518.

MINOR COMMENTS 

Thank you very much for the minor comments, as well! 

1. Should line 187 say 17-31 MIL 2x50bp? Correct typo if it is one. ; line 213

2. The figure titles and captions need to be improved. The first sentence should summarize the point of the figure (ex: in Fig 2 this could just be “Volcano plots of significant DEGs in different comparisons”), and the following unbolded sentences should describe the different subparts of the figure, describing sample types, color coding, etc. Currently, the figure titles are what should be the caption.

This was changed throughout the manuscript. 

3. In Figures 2+5, it would be beneficial to label the genes of interest that are highlighted in Figures 4+7

Unfortunately these figures are produced by a software where the labeling of genes is not possible. 

4. Line 452-453 is a double negative. Should be “did not change…either of the r-allergens….”; line 489

5. Line 460: activate; rewritten, now line 498

6. Line 556: artifacts ; line 615

7. Line 556: outlier ; line 616

---

## [Decision Letter · Decision Letter 1]

15 Feb 2022

PONE-D-21-13837R1Equine keratinocytes in the pathogenesis of insect bite hypersensitivity: just another brick in the wall?PLOS ONE

Dear Dr. Marti,

Thank you for submitting your manuscript to PLOS ONE. After careful consideration, we feel that it has merit but does not fully meet PLOS ONE’s publication criteria as it currently stands. Therefore, we invite you to submit a revised version of the manuscript that addresses the points raised during the review process.

Reviewer #2 has some additional valid comments that need to be addressed.

We look forward to receiving your revised manuscript.

Kind regards,

Angel Abuelo, DVM, MRes, MSc, PhD, DABVP (Dairy), DECBHM

Academic Editor

PLOS ONE

Journal Requirements:

Reviewers' comments:

Reviewer's Responses to Questions

**Comments to the Author**

1. If the authors have adequately addressed your comments raised in a previous round of review and you feel that this manuscript is now acceptable for publication, you may indicate that here to bypass the “Comments to the Author” section, enter your conflict of interest statement in the “Confidential to Editor” section, and submit your "Accept" recommendation.

Reviewer #1: All comments have been addressed

Reviewer #2: (No Response)

2. Is the manuscript technically sound, and do the data support the conclusions?

Reviewer #1: Yes

Reviewer #2: Yes

3. Has the statistical analysis been performed appropriately and rigorously? 

Reviewer #1: Yes

Reviewer #2: Yes

4. Have the authors made all data underlying the findings in their manuscript fully available?

Reviewer #1: Yes

Reviewer #2: Yes

5. Is the manuscript presented in an intelligible fashion and written in standard English?

Reviewer #1: Yes

Reviewer #2: Yes

6. Review Comments to the Author

Reviewer #1: All comments have been adequately addressed, points that could not be answered or investigated have been sufficiently discussed as limitations of the study.

Reviewer #2: The authors have addressed many of the prior concerns and revisions to their manuscript describing the role of equine keratinocytes in insect bite hypersensitivity. Thank you. They also describe the importance of cytokine signaling in developing and exacerbating the allergic response. Most of my revisions are minor grammatical corrections. I do have some major revisions regarding that should be addressed before acceptance. Please see my recommendations below.

Major revisions

1) Lines 42-44 and 620-621. These two sentences need to be changed to better describe the identity of keratinocytes. While the data support that keratinocytes are involved in the first line of immune defense, it is incorrect to state that they are “bona fide innate immune cells”. Expression of immune related genes shows that they are involved in the immune response, but calling them “innate immune cells” would mean that they are hematopoietically derived, which is not shown with the data and cannot be based on gene expression alone. Further characterization of cell ontogeny would be needed to call them this. Also, epidermal cells are involved in the immune response even though they are not immune cells.

Please modify these sentences in the abstract and discussion/conclusion to better describe the identity and role of keratinocytes in the immune response. Please remove “bona fide innate immune cells”.

Here is an example for the abstract: “Our data suggests that equine keratinocytes contribute to the innate immune response and are able to elicit responses to different stimuli, possibly playing a role in the pathogenesis of IBH.”

2) Figures 5 and 8 are both tables and I recommend that they be turned into tables instead of figures. Below are a few specific changes that should be made:

-Line 332: Please fix the title to read more clearly. I recommend this, or similar:

“DEGs are classified by gene families that influence (A) the immune response and (B) epithelial barrier formation and maintenance.”

-Line 334-340 and 438-441: Please add to the legend more details about the experiment that was done. In the first sentence, please add that the cell samples were analyzed by RNA-seq and that gene expression was compared between IBH-KER and H-KER. This will be redundant with the figure/table text but will greatly enhance clarity and accurate interpretation.

-Line 334-340: I recommend re-stating in the legend (not just the title) that (A) shows upregulated immune response genes and (B) shows upregulated epithelial barrier genes. Alternatively, you could combine the table into one table instead of two.

3) It would be helpful to validate the RNAseq results with qPCR analysis of lesional and/or nonlesional skin. This could be shown in new Figures 5+8 if the current figures become Tables. Please consider adding a graph that shows the gene expression of key genes (IL31, KRT80, etc) in the different conditions.

Minor revisions

Line 92: Add period/punctuation after milieu.

Line 93: Introduce ACM abbreviation here and use throughout.

Line 101: Fix grammar to say “consisting of A combination of recombinant...”

Line 102, 104: Capitalize “Type I...”. Please make sure this is correctly capitalized throughout manuscript.

Tables 2+3: Some of the term names are cut off. Please adjust so that they fit in the table, or add full term name as a table footnote.

Lines 320 + 355: Section titles are both listed as “A)”. Please fix to say “a” and “b”, or remove lettering list.

Lines 324-325: Chemokine genes do not have a dash in them, according to nomenclature rules. Please fix here and also make sure they are correctly named throughout. For example, chemokine gene names should read as “CCL20”.

Line 327: Please add “, the” after IBH-KER so that the sentence here is more clear:

“Interestingly, in IBH-KER, the atopic cytokine...”

Line 409: Fix grammar to say “belonging to the DNA replication process...”

7. PLOS authors have the option to publish the peer review history of their article (what does this mean?). If published, this will include your full peer review and any attached files.

Reviewer #1: No

Reviewer #2: No

---

## [Author Response · Author response to Decision Letter 1]

15 Mar 2022

Dear Editor, dear Reviewers,

Thank you for you useful comments.

We have corrected the manuscript according to the reviewer's comments. Changes are marked in light blue in the revised manuscript with track changes. 

There is just one comment (major revision no 3) that we cannot address. We think that validation of the RNAseq results with qPCR analysis of lesional and/or nonlesional skin is beyond the scope of this manuscript, as we have studied the response of primary keratinocyte cultures derived from skin of IBH-affected and healthy horses following in vitro stimulations. Furthermore, a recent study showed by qPCR that IL-31 was detectable exclusively in the lesional skin of IBH-affected horses, but not detectable in the skin of healthy horses (Olomski et al., 2020). This study is cited in our manuscript (Ref no 37). Additionally, we are not able to perform further experiment because the first author of our paper is no longer working in our group and our technician is unfortunately on a longer-term sick leave. Therefore, at the moment, it would be impossible for us to perform qPCRs or other experiments. We thus hope that our manuscript will be acceptable for publication without these additional experiments. 

We have made no changes to the financial and we have made no changes to the references in this revision of the manuscript. 

Reviewers' comments:

Reviewer's Responses to Questions 

Comments to the Author

1. If the authors have adequately addressed your comments raised in a previous round of review and you feel that this manuscript is now acceptable for publication, you may indicate that here to bypass the “Comments to the Author” section, enter your conflict of interest statement in the “Confidential to Editor” section, and submit your "Accept" recommendation.

Reviewer #1: All comments have been addressed

Reviewer #2: (No Response)

2. Is the manuscript technically sound, and do the data support the conclusions?

Reviewer #1: Yes

Reviewer #2: Yes

3. Has the statistical analysis been performed appropriately and rigorously? 

Reviewer #1: Yes

Reviewer #2: Yes

4. Have the authors made all data underlying the findings in their manuscript fully available?

Reviewer #1: Yes

Reviewer #2: Yes

5. Is the manuscript presented in an intelligible fashion and written in standard English?

Reviewer #1: Yes

Reviewer #2: Yes

6. Review Comments to the Author

Reviewer #1: All comments have been adequately addressed, points that could not be answered or investigated have been sufficiently discussed as limitations of the study.

Reviewer #2: The authors have addressed many of the prior concerns and revisions to their manuscript describing the role of equine keratinocytes in insect bite hypersensitivity. Thank you. They also describe the importance of cytokine signaling in developing and exacerbating the allergic response. Most of my revisions are minor grammatical corrections. I do have some major revisions regarding that should be addressed before acceptance. Please see my recommendations below.

Major revisions

1) Lines 42-44 and 620-621. These two sentences need to be changed to better describe the identity of keratinocytes. While the data support that keratinocytes are involved in the first line of immune defense, it is incorrect to state that they are “bona fide innate immune cells”. Expression of immune related genes shows that they are involved in the immune response, but calling them “innate immune cells” would mean that they are hematopoietically derived, which is not shown with the data and cannot be based on gene expression alone. Further characterization of cell ontogeny would be needed to call them this. Also, epidermal cells are involved in the immune response even though they are not immune cells.

Please modify these sentences in the abstract and discussion/conclusion to better describe the identity and role of keratinocytes in the immune response. Please remove “bona fide innate immune cells”.

Here is an example for the abstract: “Our data suggests that equine keratinocytes contribute to the innate immune response and are able to elicit responses to different stimuli, possibly playing a role in the pathogenesis of IBH.”

Thank you very much for your comprehensive insight. The changes were adapted in the lines 45-47 and 617-618. 

2) Figures 5 and 8 are both tables and I recommend that they be turned into tables instead of figures. Below are a few specific changes that should be made:

Thank you for your suggestion. The figures are now turned into tables. 

-Line 332: Please fix the title to read more clearly. I recommend this, or similar:

“DEGs are classified by gene families that influence (A) the immune response and (B) epithelial barrier formation and maintenance.”

Thank you very much. Title is now fixed in 332-333. 

-Line 334-340 and 438-441: Please add to the legend more details about the experiment that was done. In the first sentence, please add that the cell samples were analyzed by RNA-seq and that gene expression was compared between IBH-KER and H-KER. This will be redundant with the figure/table text but will greatly enhance clarity and accurate interpretation.

Thank you. The suggested changes are now implemented in lines 333-336 and 444--448. 

The line 332 and 334-3340 and 

-Line 334-340: I recommend re-stating in the legend (not just the title) that (A) shows upregulated immune response genes and (B) shows upregulated epithelial barrier genes. Alternatively, you could combine the table into one table instead of two.

Thank you! The changes are implemented in the lines 336-337.

3) It would be helpful to validate the RNAseq results with qPCR analysis of lesional and/or nonlesional skin. This could be shown in new Figures 5+8 if the current figures become Tables. Please consider adding a graph that shows the gene expression of key genes (IL31, KRT80, etc) in the different conditions.

In this manuscript we have investigated the response of primary keratinocytes derived from skin of IBH-affected and healthy horses following in vitro stimulations. We find that validating our findings in lesional and nonlesional skin is thus beyond the scope of the study presented here. Furthermore, a recent study showed by qPCR that IL-31 was detectable exclusively in the lesional skin of IBH-affected horses, but not detectable in the skin of healthy horses (Olomski et al., 2020). This study is cited in our manuscript (Ref no 37). 

Additionally, we are not able to perform further experiment because the first author of our paper is no longer working in our group and our technician is unfortunately on a longer-term sick leave. Therefore, at the moment, it would be impossible for us to perform qPCRs or other experiments. 

Minor revisions

Thank you very much. All the minor revisions are implemented in the following lines: 

Line 92: Add period/punctuation after milieu. Line 92.

Line 93: Introduce ACM abbreviation here and use throughout. Line 93 and all throughout manuscript. 

Line 101: Fix grammar to say “consisting of A combination of recombinant...”. Line 101. 

Line 102, 104: Capitalize “Type I...”. Please make sure this is correctly capitalized throughout manuscript. Lines 102, 104 and all throughout manuscript. 

Tables 2+3: Some of the term names are cut off. Please adjust so that they fit in the table, or add full term name as a table footnote. Implemented in both tables. 

Lines 320 + 355: Section titles are both listed as “A)”. Please fix to say “a” and “b”, or remove lettering list.

Implemented in lines 320 (a) and 361 (b). 

Lines 324-325: Chemokine genes do not have a dash in them, according to nomenclature rules. Please fix here and also make sure they are correctly named throughout. For example, chemokine gene names should read as “CCL20”. Lines 324-325 and throughout manuscript. 

Line 327: Please add “, the” after IBH-KER so that the sentence here is more clear:

“Interestingly, in IBH-KER, the atopic cytokine...” Line 327.

Line 409: Fix grammar to say “belonging to the DNA replication process...” Line 415.

---

## [Editor Report · Decision Letter 2]

18 Mar 2022

Equine keratinocytes in the pathogenesis of insect bite hypersensitivity: just another brick in the wall?

PONE-D-21-13837R2

Dear Dr. Marti,

We’re pleased to inform you that your manuscript has been judged scientifically suitable for publication and will be formally accepted for publication once it meets all outstanding technical requirements.

Kind regards,

Angel Abuelo, DVM, MRes, MSc, PhD, DABVP (Dairy), DECBHM

Academic Editor

PLOS ONE
---

## [Editor Report · Acceptance letter]

27 Jun 2022

PONE-D-21-13837R2 

Equine keratinocytes in the pathogenesis of insect bite hypersensitivity: just another brick in the wall? 

Dear Dr. Marti:

I'm pleased to inform you that your manuscript has been deemed suitable for publication in PLOS ONE. Congratulations! Your manuscript is now with our production department. 

Kind regards, 

on behalf of

Dr. Angel Abuelo 

Academic Editor

PLOS ONE